# Explaining Religious Revival in the Context of Long-Term Secularization

**Jörg Stolz** [1,*] and **David Voas** [2]

1   Institut de sciences sociales des religions, Université de Lausanne, 1015 Lausanne, Switzerland
2   Social Research Institute, University College London, London WC1E 6BT, UK
*   Correspondence: joerg.stolz@unil.ch

**Abstract:** Secularization theory has often been criticized for not being able to explain counterexamples. However, secularization theorists argue that transitory religious resurgences are expected to occur even in modernizing conditions. The aim of this article is to identify mechanisms that can explain the temporary upswing of religion against the backdrop of long-term modernization. We classify the mechanisms under five broad headings: crisis, reaction, transition, state intervention, and composition. Historical examples are provided to illustrate these mechanisms. The mechanisms are not mutually exclusive and can be understood within the framework of rational action.

**Keywords:** religious revival; secularization; religious change; social mechanism





## 1. Introduction

Secularization theory has often been criticized for the existence of counterexamples that appear to contradict the theory. Critics of secularization theory argue that it is difficult to accept the theory when there are cases where modernization and religious stability or even resurgence coexist. They question how secularization theory can account for countries such as the United States, Brazil, India, South Korea, Russia, Georgia, Ghana, Eritrea, Iran, Turkey, or Egypt (to name a few) (Berger et al. 2008; Casanova 1994; Joas 2017).

In a well-known paper, Wallis and Bruce (1995, p. 702) proposed an answer to the question of how modernization can coexist with religious stability or resurgence. They argued that "[modernization generates] secularization except where religion finds or retains work to do other than connecting individuals to the supernatural." This "other work," Wallis and Bruce continued, comes in two forms: "cultural defense" and "cultural transition". In the former case, religion serves as an identity resource to unite a group against an outside threat. In the latter case, religion is used by immigrants to bond and find strength in a host country. According to the authors, these mechanisms are compatible with ongoing modernization and are transitional since religiosity drops once the outside threat subsides or the immigrants have been assimilated.

We concur with the argument put forth by Wallis and Bruce, but we also believe that their typology of mechanisms should be refined and expanded. Our central question is thus: How can secularization theory account for counterexamples, specifically the coexistence of religious resurgence alongside modernization?

The new typology we propose maintains the two types proposed by Wallis and Bruce but renames them "reaction" (formerly cultural defense) and "transition" (formerly cultural transition). We add three new types: "crisis," "state intervention," and "composition". Additionally, we argue that the different mechanisms can be integrated into a broader framework of rational action.[1]

The main focus of this paper is to provide a typology of mechanisms that can account for purported counterexamples to secularization theory: cases where religion persists or revival coincides with long-term modernization. We are therefore not concerned with

cases where religion remains stable in the absence of modernization. Societies such as the Old Order Amish in the United States or the Yanomami in Venezuela and Brazil have not modernized and thus are not expected to secularize (Kephart and Zellner 1988; Dawson 2009). African countries with a low Human Development Index, such as South Sudan, Chad, Niger, and Mali, have not undergone sufficient modernization to result in observable secularization.

There is a clear problem with trying to explain exceptions within the context of a theory. Scholars may attempt to "patch up" their theory to deal with apparent counterexamples. Kuhn ([1962] 1970) famously argued that scholars often try to "explain away" anomalies in order to save their theories. Although this is a valid concern, it does not mean that we should not attempt to examine and explain anomalies. Science has often made significant progress when theories have been developed to try to make sense of what seemed inexplicable. To avoid the perception that what follows is an "immunizing strategy," we will discuss how the proposed mechanisms can be used in empirical research and thus be falsified.

## 2. Theory: Modernization, Secularization, and Religious Revival

To set the stage for our new typology, a few definitions are in order. At its core, secularization theory states that modernization leads to secularization. All other specifics are controversial. In this section, we will define and briefly discuss the terms modernization, secularization, and religious revival.

We define secularization as the decline of the importance of religion and religiosity in a country or region on a societal, organizational, and individual level (Dobbelaere 2002; Pollack and Rosta 2017; Stolz 2020b; Bréchon 2013). Secularization can be seen, for example, in the declining presence of religion as a guiding principle in the constitution and the laws of a state, in the shrinking extent to which religious groups interfere in the social practices of a society (e.g., education, health provision, science, politics), and in a decline in religiosity, such as religious belief, practice, and affiliation of individuals (Zuckerman 2008).[2]

In this paper, we address the specific empirical criticisms of secularization theory linked to cases that appear to contradict the theory. However, there are also important conceptual criticisms. For instance, some scholars believe that secularization theory is based on an outdated evolutionist perspective, implying that non-western countries have to "catch up" to western countries in terms of modernization and secularization. What is needed instead, these authors claim, is a gendered and decolonial narrative of religious change (Müller 2020) or one that is more attentive to qualitative changes linked to the erosion of the nation-state (Gauthier 2020). Limitations of space prevent us from entering into these discussions here (see, for example, Gauthier 2020; Müller 2020; Voas 2020; Wilkins-Laflamme 2020; Stolz 2020a)

Modernization can be defined as the process that leads societies to a greater level of complexity in at least three domains: technological, institutional, and cultural (Stolz and Tanner 2019; Ruiter and Van Tubergen 2010).[3] Technological modernization derives from scientific progress that results in innovations and technical know-how, creating a world increasingly made and controlled by human beings. This world is giving rise to an unprecedented increase in income, allowing people to use all the new instruments of problem solving. Examples include the discovery of penicillin, electricity, bacteria, X-rays, or the invention of clocks, airplanes, computers, and the internet. Institutional modernization concerns the change of regulations and social organization leading to the differentiation of societal sub-spheres, democratization, and bureaucratization. It is related to a process in which social activities are classified into functional systems (e.g., business, education, law, politics, and medicine). It leads to a situation in which decisions at different levels of social organization are increasingly taken and legitimized by the voices of the participants, and state organizations increasingly function according to formal rules. Cultural modernization refers to a social evolution that accepts the ideas

of the Enlightenment and a scientific vision of the world; it includes the development of individualistic, emancipatory, egalitarian, and democratic values.

Why should modernization lead to secularization, according to secularization theory? Many mechanisms have been proposed in the literature, and since this is not the primary focus of the present paper, we point readers to the literature (Pollack and Rosta 2017; Voas et al. 2013; Bruce 2011; Bréchon 2018). On a very general level, we can say that modernization creates non-religious solutions to the problems of human life that seem more efficient than those offered by religion (Stolz and Tanner 2017; Stolz et al. 2016). Individuals and groups will therefore, in the long term, substitute secular options for religious options.

Thus, technological modernization allows humans to solve their daily problems in a technical and non-religious way. A serious illness, for example, can be treated technically and medically and does not require prayer or exorcism to be cured. Likewise, institutional modernization, which differentiates social spheres, makes non-religious spheres independent of religion and therefore reduces the importance of religion in these areas. In addition, the state can impose regulations and institutions to ensure religious peace. For example, the Edict of Nantes in 1598 demanded tolerance towards Protestants and granted them full citizenship in a Catholic France. Finally, cultural modernization leads to the decline of the religious because it points to new non-religious causes for phenomena that religion explained supernaturally (e.g., natural disasters or diseases). Cultural modernization paved the way for a scientific worldview, ready—at least in principle—to question all elements of the received worldview and hostile to claims of transcendent truth. Combined with growing situations of cultural diversity, this engenders a situation in which the elites, then the masses, experience the relativization of their consciousness and believe with increasing difficulty in "transcendental powers" such as gods, angels, and devils. To recap, in all three areas of modernization—technological, institutional, and cultural—secular options are created that tend to "crowd out" religious ones.

We define a religious revival (or resurgence) as an increase in the importance of religion on the individual, organizational, and societal levels in aggregate in a given country or region. We use revival as a religiously neutral term regardless of whether or not the religion itself regards a phenomenon as a "revival".

A counterexample to the theory of secularization is a country or region that is undergoing modernization but is not experiencing secularization or is instead experiencing a resurgence of religion.

### 3. Mechanisms of Revival[4]

Expanding the Wallis/Bruce typology, we distinguish five broad types of mechanisms: crisis, reaction, transition, state intervention, and composition (Figure 1). The first mechanism ("crisis") differs from the four others in that it is just the temporary reversal of the central modernization-secularization mechanism of the basic theory. The religious revival is explained by a temporary de-modernization. The other four mechanisms, however, point to additional causal pathways that allow for religious revival even though modernization is occurring.

#### 3.1. Crisis

A religious revival is possible if a societal crisis is so deep that we may speak of a temporary "de-modernization"(Rabkin and Minakov 2018).[5] In such cases, we should find one or a combination of several of the following phenomena: the de-differentiation of societal subsystems, a breakdown of the welfare state, an abandonment of democracy, a declining level of education and longevity, and a steeply declining level of GDP/capita. It is also possible that societies may show de-modernization in some areas but modernization in others. Major crises can destabilize material existence and ways of interpreting the world, leading to extreme inequalities and insecurities. In these situations, religious groups may provide a sense of security and help reintegrate people.

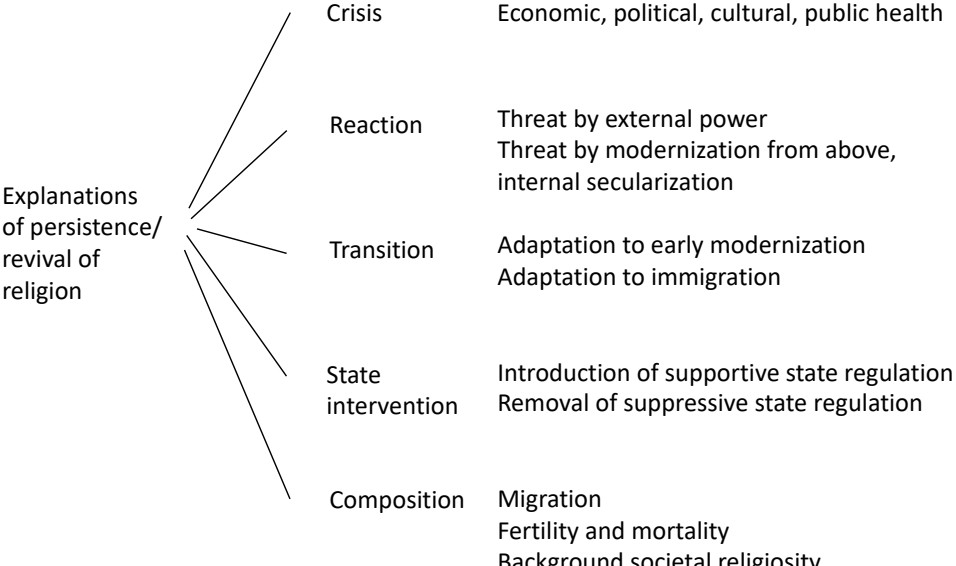

**Figure 1.** A typology of mechanisms of persistence and revival of the religious.

An excellent example of religious revival following a societal crisis can be observed in post-Communist Orthodox countries, such as Armenia, Belarus, Georgia, Moldova, Romania, Russia, Serbia, and Ukraine, after the disintegration of the Soviet Union around 1990 (Evans and Northmore-Ball 2012; Pickel 2009; Voicu and Constantin 2012; Pew Research Center 2017).[6]

In Russia, for example, membership in the Orthodox Church increased from 31% in 1991 to 72% in 2008 (Pew Research Center 2017). Belief in God and self-identification as religious also increased, but to a lesser extent. However, attendance at church services remained at a low level. Pollack and Rosta (2017) interpret these developments as a resurgence of religious identity and nationalism. They argue that the collapse of the Soviet Union in 1991 represented a significant crisis, both economically and in terms of identity. The sudden loss of a shared social identity and cultural project was filled by the idea that being Russian equated to being Orthodox, and this could be a source of pride. In this way, the Orthodox Church provided legitimacy for the state, and the state supported the church through legal and financial means. As an example, Putin has publicly aligned himself with Patriarch Kirill and even stated in 2007 that traditional religions were as crucial for Russia's security as nuclear defense (Pollack and Rosta 2017, p. 154).

The same phenomenon can be observed even more clearly in Georgia (Jödicke 2015; Köksal et al. 2019) (Figure 2). According to Stolz et al. (2023), the religious revival began slowly around 1985 and gained momentum in 1990. This is evident in the increase in the number of churches, religious affiliation, perceived importance of religion, and attendance at religious services. As in Russia, the resurgence of religion in Georgia was largely influenced by the severe economic and political crisis that ensued after the collapse of the Soviet Union. The country faced hyperinflation, war, and civil war, a significant increase in crime and corruption, and a loss of approximately 70% of its economy in a short period of time. In this situation, people did not have confidence in the government, parliament, or political parties. Instead, they trusted the one institution that appeared to be functioning normally: the Orthodox Church, led by its charismatic patriarch Ilia II. Similar to Russia, the church provided a sense of identity to Georgians and legitimized political leaders who aligned themselves publicly with the patriarch.

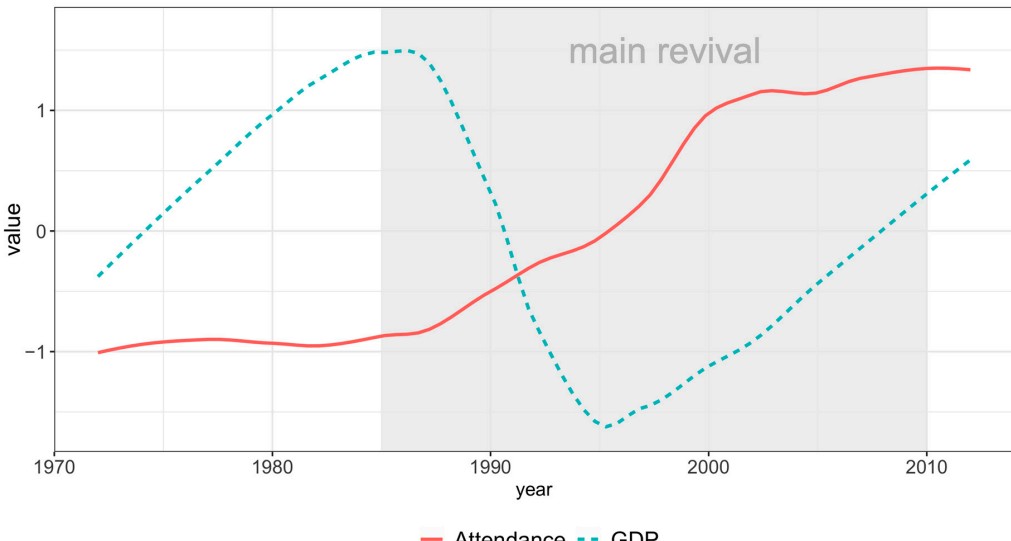

**Figure 2.** Child attendance and GDP per capita in Georgia, 1972–2012. Note: GDP per capita in USD, scaled; percentage of individuals with at least monthly attendance as a child, scaled; attendance is an eight-step variable ranging from eight (more than once a week) to one (never); shaded area indicates time of strong religious revival. Source: Stolz et al. (2023).

Some studies have found evidence of a link between war experiences and religiosity. Henrich et al. (2019) conducted a quasi-experiment and found that higher exposure to war in Uganda, Sierra Leone, and Tajikistan led to a significantly higher probability of belonging to a religious group, even years after the conflict. Additionally, Pew Research Center (2018, p. 40) reports that civil wars in Liberia, Rwanda, and Ghana seem to have led to some form of religious revival, as evidenced by the fact that younger individuals in these countries were more religious than older individuals.

Other studies have found that earthquakes can also elicit higher religiosity. In a worldwide study, Sinding Bentzen (2019) states that individuals tend to become more religious when they have been affected by an earthquake or other unpredictable disaster. The study also found that while the main effect tends to subside after a number of years, there is a residual effect that is transmitted to the next generation. Another study, conducted by Sibley and Bulbulia (2012), looked at the effect of the earthquake in Christchurch, New Zealand, on 23 February 2011, and reported that individuals living in the affected area significantly increased their identification with a religious or spiritual group compared with individuals who did not live in that area.

It is important to note that not all studies have found significant effects of war and the economic crisis on religiosity. For example, Bruce and Voas (2016) claim that major crises such as World War I and II and the Great Depression did not result in an increase in religiosity in Great Britain. Additionally, the Pew Research Center (2018, p. 40) reported that in countries experiencing significant conflicts, such as Bosnia-Herzegovina, the Democratic Republic of the Congo, and the Palestinian territories, there was no evidence of a religious revival as measured by the gap in religiosity between older and younger individuals.

In some examples, such as the Iranian revolution, it is challenging to establish causality. In this case, de-modernization likely contributed to the resurgence of religion, but it is also likely that religious forces actively sought to de-modernize society (Axworthy 2019).

*3.2. Reaction*

A religious revival is also possible if a group is threatened from outside, from above by an elite, or from some cultural change. In such a situation, religion may be used by the threatened actors as an identity resource to coordinate their fight against the perceived aggressor and to build up and defend a positive group identity. This strategy was called "cultural defense" by Wallis and Bruce (1995, p. 702).

A well-known example of religious persistence and even growth during a period of suppression is Poland during the communist era (Borowik 2002; Pollack and Rosta 2017). For the predominantly Catholic Polish society, Catholicism had long been an important aspect of Polish identity and helped sustain the society during previous occupations by various powers. This remained true during the Soviet period, as communism and state atheism were perceived as foreign threats against which a Polish–Catholic identity was reinforced. Eberts (2007, p. 817) writes: "In the most recent past, for the majority of Poles, the Church has been a bastion of freedom and a source of protection from and opposition to the communist authorities. (...) It emerged from the communist period not only as the highest moral authority but also, in large measure by securing important concessions from the communist authorities, as the most powerful institution in the country."

Another example is the Catholic revival in Ireland as a reaction to British colonialism in the 19th century (White 2010). The Irish people responded both to British political domination and to attempts by Protestants to convert Irish Catholics. In doing so, they merged Catholic and Irish identities to form an ideology that legitimized their resistance. This process culminated in the Irish War of Independence from 1919 to 1921, which ultimately led to the independence of Ireland. The counterpart of the Catholic revival in Ireland was the Protestant Ulster revival in Northern Ireland in 1859. This revival was sparked by an evangelical revival in the United States and had much to do with the fact that evangelicals were expecting and preparing for such a revival (Holmes 2005). However, in Northern Ireland, it acquired an additional force by uniting different fractions of Protestantism against Catholics, who were seen as an outside, threatening force controlled by the Vatican.

The Islamic revival in predominantly Muslim countries since the 1970s is another case in point. The emergence of the Muslim Brotherhood and the Islamic revival in Egypt, the Iranian Revolution, or the re-Islamization of Turkey are all clearly defensive reactions to strong and rapid modernization from above (Ahmed 2011; Kepel 1985; Axworthy 2019; Rabasa and Larrabee 2008).[7] Modernization and secularization were symbolized, among other things, by new products, the education of women, and especially the removal of the veil for women. The Muslim reaction movements criticized these modernizing tendencies, which began in the 1920s but experienced a significant increase in the 1970s (Lapidus 1997). They worked tirelessly, and ultimately quite successfully, to counteract modernizing tendencies that were branded as "western" and contrary to Islam. One of the most notable actions seen in all these countries was the reinstatement of the veil for women as a symbol of legitimacy.[8]

The Islamic revival is a reaction not only against rapid modernization but also against colonization by the West and the foundation of the state of Israel. Thus, following the battle of Tel el-Kebir, Egypt became a de facto British protectorate in 1882, a situation that ended only with the 1954 revolution. The long de facto occupation by the British and the defeat in the battle of 1967 against Israel were seen as humiliating by many (Carvalho 2009, p. 13).[9] The Muslim Brotherhood, founded by Hassan al-Banna in 1928, criticized from the start both the British occupation of Egypt and the continuing influx of Jews into Israel. It preached a return to Muslim roots and a pure Islam that would restore both Muslim pride and strength (Ahmed 2011; Kepel 1985).[10]

Israel is another case where elevated religiosity is arguably explained in part as a reaction to an outside threat. Younger Israelis are, on aggregate, not more secular than older ones, and the percentage of Orthodox Jews over time has risen between 2002 and 2013 (Pew Research Center 2016, pp. 20, 36). Israel is in a state of latent conflict with surrounding states and is internally challenged by divisions between secular and religious Jews as well as by tensions with the Arab minority and conflicts with Palestinians (Pew Research Center 2016). In this situation, religious and ethnic Jewish identities permit a tightening of group cohesion and solidarity. However, high average religiosity in Israel is also a result of composition effects: high fertility among the very religious Haredi Jews as well as the immigration of observant Jews.

Interestingly, the perceived threat may also arise from cultural change within the group in the form of internal secularization (Stark and Bainbridge 1985). Strictness and commitment may fade, along with the weakening of beliefs, practices, and identity boundaries. Sometimes a charismatic founder dies, a second generation arrives, and members achieve financial success and seek acceptance in wider society by becoming less "different". John Wesley already thought that Methodism led to riches, which in turn would lead to a weakening of the faith: "... the Methodists in every place grow diligent and frugal; consequently, they increase in goods. Hence, they proportionately increase in pride, in anger, in the desire of the flesh, the desire of the eyes, and the pride of life. So, although the form of religion remains, the spirit is swiftly vanishing away." (Wesley 1786, cited by Johnson (1976, p. 368)).

When such an internal secularization happens, reactionary movements are likely. Leaders emerge, claiming that the community has strayed from the right path and that it should go back to a former, purer, more authentic way of practicing and believing. Methodism itself emerged in England as such a response. To different degrees, the Reformation in 16th-century Europe, the Christian Fundamentalist movement in the United States in the early 20th century, the rise of the "moral majority" in the 1980s in the United States, or the Salafi movement in Islam in the 19th century are examples of reactions to (at least in part) internal secularization (Lapidus 1997; Riesebrodt 2000; Casanova 1994).

*3.3. Transition*

Religious resurgence may also appear when groups of individuals choose to adapt to societal change. In this case, it may be used not so much defensively as in the previous type but rather to legitimize new values and techniques of life and, in this way, smooth the transition. Furthermore, early modernization may have given religious groups new and powerful techniques to get their message across and find new members. This type of mechanism was labeled "cultural transition" by Wallis and Bruce (1995, p. 702). They had mainly the smoothing function of religion for immigrants in mind, but we believe that a transition function of religion can also be found in many early stages of modernization.

3.3.1. Adaptation to Early Modernization

Often, we seem to find transitory resurgences of religion in the early stages of modernization and urbanization. Early modernization and urbanization make life increasingly unpredictable. Individuals may be forced to work in factories, may face rapidly moving salaries and prices, and may have to move to the cities to find work. They may fear losing their frameworks of interpretation. In such a situation, many people are looking for new ideologies, values, and social ties that give them new security and, at the same time, equip them for the new societal contexts.

An example of this is the rise of Methodism in England during the 18th and early 19th centuries, where strong modernization and urbanization led individuals to become receptive to an ideological message that provided them with new guidance (Bruce 2011, p. 186; Luker 1988; Hempton 2005). Methodism, while more individualistic than traditional Anglicanism, fit better with modern society but also worked to strengthen the solidarity and restore faith of individuals in a traditional order where, supposedly, individuals received what they deserved. The rapid expansion of Pentecostalism in Brazil and Sub-Saharan Africa can also be explained by this approach (Martin 1990; Pew Research Center 2013). Pentecostalism offers a more individualistic approach to living one's faith compared with Catholicism while also reintegrating individuals who have been affected by modernizing forces into structured communities of believers, similar to Methodism in the past.

3.3.2. Adaptation to Immigration

Religion may also smooth the transition to a new societal context for immigrants. Religious groups may help immigrants both preserve religious and cultural traditions and insert them into new social networks (Hirschman 2004). As a result, immigrants may find

themselves more religious in the new host country than they had been in the emigration context. It has often been claimed that the high religiosity in the United States was at least in part caused by the fact that religious groups played a key role in the integration of immigrants into American society (Warner and Wittner 1998).

*3.4. State Intervention*

3.4.1. Introduction of Supportive State Regulation

Transitory religious resurgences are also possible as a result of state intervention. We can distinguish cases where state regulation supports, promotes, or even requires religious adherence from cases where the state removes obstacles to belief and practice. In the former case, a state lacking legitimacy will actively support religion as a legitimation resource to sustain political power, thus creating a "religious revival from above". Such a state may offer both legal advantages and financial resources to one or several religions close to the state, and it may at the same time suppress other religions (Fox 2015). For example, the first phase of the Franco regime in Spain relied on Catholicism as an important source of legitimation and granted it significant legal and financial advantages (Pastor 2007).

A more recent example is the spectacular growth of Buddhism, Protestantism, and "Other Religion" in South Korea in the 1970s, which was strongly guided by the state (Lee and Suh 2017). General Park Chung Hee, who came to power in 1961 by coup d'état, modernized society "from above" in a precipitous and authoritarian manner. Urbanization was very rapid. At the same time, the generals suppressed the dominant folk religions and strongly encouraged individuals to join religions seen as beneficial to the state, in particular Buddhism and Protestantism. Many people struggled to adapt to these abrupt changes, and a substantial proportion translated their old folk religion into Buddhism, while others switched to Protestantism, considered particularly modern. The return of the religious is therefore, according to Lee and Suh, mainly to be seen as the result of action by the state that was in the process of being created. South Korea has since been democratized and has become more politically stable; as a result, there are obvious signs of secularization. The situation seems increasingly similar to western democracies.

The pro-Islamic policies of the AKP in Turkey under Recep Tayyip Erdogan since 2002 are another example of state intervention. The government increased funding for the religious sector, incorporated religious groups into the party and the state organization, promoted the construction of new mosques, and encouraged women to wear the veil (Aksoy and Billari 2014; Cevik 2019; Rabasa and Larrabee 2008). While the revival in Turkey has many of the same roots as the Islamic revival in other countries, supportive state regulation certainly helped the resurgence of religion.

3.4.2. Removal of Suppressive State Regulation

Another possibility is that the state removes suppressive regulation, thus allowing religion to rebound to a level consistent with expectations (Stark and Bainbridge 1989; Inglehart and Welzel 2005). Negative sanctions on some or all religious groups can include administrative obstacles to religious activity, the spreading of negative information about leaders and/or members of religious groups, obstructing the careers of members or their children, or even interdictions on religious practice. Sometimes a particular group is privileged as the country's "established" religion. Even the removal of such recognition could theoretically lead to a religious upswing.[11] It seems clear that removing official obstacles could lead to a boost in religious participation.

There are different theoretical positions on the question of how much religiosity will increase once official restrictions are lifted. One argument is that religiosity is a fixed trait and religious demand is stable across societies, which implies that allowing all groups to be free to organize and recruit would produce societies with uniformly high levels of religious involvement (assuming that the supply is sufficient) (Stark and Finke 2000). Another position is that a society's level of aggregate religiosity depends on the degree to which it has modernized, and if the state has repressed religious expression below this

natural level, the removal of restrictions will generate a revival that takes it back to the appropriate point (Inglehart 2021; Voas 2008). A third position is that the removal of state suppression of religion will lead to a rebound to a specific, possibly unknown, level of aggregate religiosity in a given country.

A good historical example of the removal of suppressive state regulation is the introduction of freedom of religion in various western countries (Hafner 2011; Johnson and Koyama 2019). In France, freedom of religion was installed after the revolution in 1789 by the Declaration of the Rights of Man and of the Citizen. In Germany, freedom of religion was granted partly in 1848 and then fully in 1919, in the "Weimarer Reichsverfassung". In Switzerland, freedom of religion was granted (albeit in a limited way) in 1848 with the establishment of the federal state. These new liberties helped the emergence of new Christian free churches and were one of the facilitating conditions of the evangelical revival in the 19th century in many of these countries (the European version of the "Second Great Awakening" in the United States). Causality runs in both directions, as freedom facilitated the emergence of dissenting churches, and the dissenters actively fought for religious liberty.

A second set of examples can be found in those parts of the former Soviet Union and eastern Europe that abolished communist suppression of religion. Research generally seems to show that there were transitory upswings in religion in most of these countries around 1990, but that traditionally Catholic countries then returned to a secularization path relatively quickly, while the religious upswing in traditionally Orthodox and Muslim countries was of much longer duration (Northmore-Ball and Evans 2014).

Figure 3 shows how official action depressed child baptism from 1950 onward in East Germany. The removal of state repression in 1990, once the two Germanies were no longer separated, led to a clear but only transitory upswing in the East (Stolz et al. 2020).

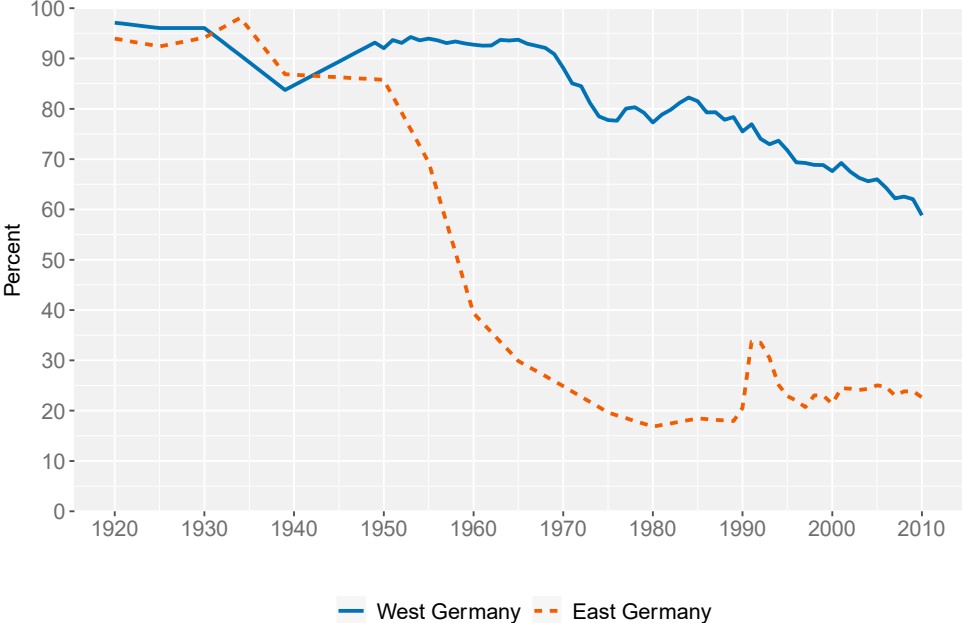

**Figure 3.** Protestant and Catholic baptisms in East and West Germany, 1920–2010. Notes: Protestant and Catholic baptisms in East and West Germany, 1920–2010. Notes: Rate of baptism is calculated as the number of Protestant and Catholic baptisms in a year divided by the total number of births in the population. Source: Stolz et al. (2020, p. 12).

*3.5. Composition*

A final type of mechanism regards composition effects. The revival is here not caused by a change in the behavior and beliefs of individuals but rather by the fact that the composition of the population is changed either by migration or fertility and mortality.

### 3.5.1. Migration

Migration may lead to religious revival through composition effects when there is immigration from more religious countries; these immigrants thus raise average religiosity in the destination country. For example, people from Orthodox countries settling in western countries are often more religious than the autochthonous population (Giordan 2018). Emigration could theoretically also result in a religious resurgence if, for some reason, secular people were more likely to leave than religious people. However, we do not know of any case where such a revival has been empirically demonstrated.

### 3.5.2. Fertility and Mortality

A society or region may also become more religious when religious groups exhibit higher fertility and/or lower mortality rates than secular groups. The Amish in America are a good example of the former. This highly religious group shows a stable fertility rate of between six and eight children per Amish woman, producing a much higher rate of natural increase than the rest of the population (Wasao et al. 2021). Similarly, the high fertility of Israel's Haredi (ultra-Orthodox) population causes their growth rate to be four percentage points higher than that of any other social group in the country (Cahaner and Malach 2021).

This mechanism is not limited to specific groups. Religion appears to be on the rise globally, as populations in poor, religious countries have comparatively high fertility and declining mortality (Norris and Inglehart [2004] 2011). The Pew Research Center (2015) estimated that the number of unaffiliated individuals will only increase slightly from 2010 to 2050, but their share of the population will decrease from 16.4% to 13.2%. Conversely, the religiously affiliated population will rise in both absolute and relative terms. As the demographic transition is a widespread phenomenon, it is unlikely that countries with high fertility will continue to grow indefinitely (Canning 2011). To that extent, the global effects of religious fertility differentials may ultimately be temporary.

### 3.5.3. Background Societal Religiosity

A third way in which composition may be relevant for religious resurgence concerns the overall level of religiosity in a given society. The more people there are who were religiously socialized, the larger the pool of potential recruits for a religious revival. Conversely, the more a society is already secularized, the more difficult it is to revive any dormant religiosity, as it has become less common. This factor could help to explain why every Christian revival (the Reformation, the sects from the era of the English Civil War, Methodism, and early 20th century Pentecostalism in the West) seems to have been smaller than the one before (Bruce 2002, p. 176). Similarly, the various historical Islamic revivals were enabled by high background levels of religiosity (Lapidus 1997). To the extent that Muslim societies secularize, future revivals should diminish in size.

## 4. Integrating the Mechanisms into a General Theoretical Scheme

Having described and illustrated the five types of mechanisms, we can now seek to integrate them into a general theoretical scheme (Figure 4).

The basic idea is that a large-scale modernization process, propelled by technological innovation and concurrently growing diversity, egalitarianism, and democratization, leads to secularization. Innovations have the effect that human life problems can be solved not just in a religious and symbolic manner but in a secular, more technical, and more efficient way. Since humans are in a very general sense assumed to be rational, that is, seeking to solve their problems in a way that is most satisfying and least costly to them, they will in the long run opt for the secular option. For example, once biomedicine becomes a powerful tool for treating illnesses in a technical way, secular suppliers (medical doctors, hospitals) tend to crowd out religious suppliers of healing. The innovation of the welfare state as well as accident, illness, and retirement insurance tends to crowd out the religious supply of insurance that was based on giving individuals hope and integrating them into a fellowship of coreligionists.

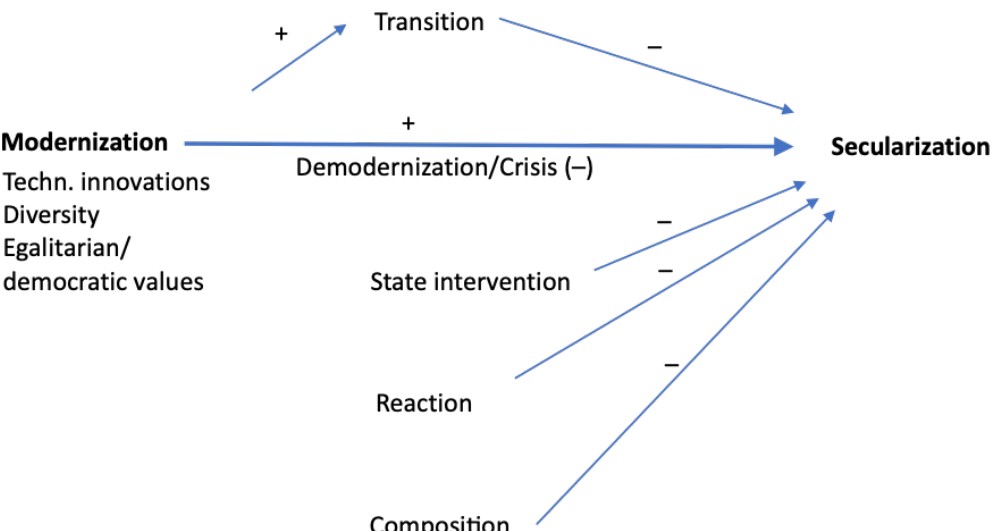

**Figure 4.** Theory of religious-secular competition.

These large modernization-secularization changes are accompanied by the transitory revivals enumerated in our paper. In the case of "transition," it is early modernization itself that leads to certain (but only transitory) religious resurgences. In the case of a crisis, it is a temporary reversal of modernization that leads to revival. In the three other cases (state intervention, reaction, and composition), additional mechanisms account for the religious resurgence.

In a long perspective, humanity seems to be on a path of continuing and accumulating innovation, replacing symbolic with technical solutions to life problems, increasing the disposable resources of families and individuals, and adapting regulations accordingly (World Bank 2021).[12] It is therefore reasonable to expect that secularization will, in the long run, prevail and continue. Correspondingly, there are data that seem to confirm that secularization is a worldwide process (Molteni 2021; Norris and Inglehart [2004] 2011; Pew Research Center 2018).

## 5. Conclusions

In this paper, we have focused on how religious persistence or revival can be explained within the framework of secularization theory. Specifically, we have discussed five types of mechanisms that can explain the persistence or revival of religion despite ongoing modernization: crisis, reaction, transition, state intervention, and composition. The contribution of the paper is theoretical: we have expanded upon the typology presented by Wallis and Bruce (1995, p. 702) and have placed the argument within a larger framework of rational action.

Several points should be noted. First, there is a common theme of using religion to bolster social cohesion and identity as a reaction to threatening forces in four of the mechanisms. The mechanisms "crisis," "reaction," "supportive state intervention," and "transition" are all about using religion in this way, although the types of threats differ. Sometimes what is threatening is a crisis, sometimes rapid social change, and sometimes an occupying power. Even "internal secularization" belongs to this list, since members of a religious group feel threatened by what they see as a dangerous weakening and dilution of the religious message.

Second, the five types are not mutually exclusive. Many cases of religious revival involve a combination of two or more of these types. A combination of several mechanisms will make a religious resurgence more likely than if only one mechanism is present. For example, the Islamic revival after 1970 can be attributed to a mix of "reaction" and "crisis". Similarly, in the Orthodox revival, a combination of "crisis," "reaction," "removal of

state suppression," and "introduction of state support" can be observed. The fact that mechanisms can appear simultaneously may make it difficult to determine causality.

Third, the mechanisms are not deterministic, and the reaction to the independent variables in our mechanisms is not necessarily religious. "Reaction" or "crisis" may lead to political mobilizations, riots, revolutions, or emigration without or with little religious resurgence involved. For the mechanism to become one of religious resurgence, additional context factors have to be given. In the Georgian revival, we find that there was an intelligent and charismatic patriarch and a protest movement that brought together both religious and political opponents of the communist regime. Against this background, the societal crisis could lead to a major religious upswing (Stolz et al. 2023). In the case of the rise of Methodism in Cornwall in the 18th century, a need for "transition" through the introduction of large-scale mining was combined with the fact that John Wesley tirelessly preached in the region. The combination of these factors had the effect that the societal reaction was a religious one (Luker 1988).

As a general caveat, it is clear that the mechanisms here proposed should not be used as a way to explain religious revival in an ad hoc manner. Such ad hoc use is problematic because it can lead to cherry-picking cases that fit a specific explanation. To avoid this type of ad hocism, several techniques can be used, including causal designs to identify causality through natural experiments, systematic comparative research, and single case studies that show how different mechanisms played out (Angrist and Pischke 2009; Della Porta 2008; George and Bennett 2005).

Typologies cannot be proven to be true or false; they can only prove to be more or less useful than alternative schemes. The effectiveness of our proposed typology must therefore be demonstrated through empirical research. We encourage studies that apply it.

**Author Contributions:** Conceptualization, J.S. and D.V.; writing—original draft preparation: J.S.; writing—review and editing: J.S. and D.V.; visualization: J.S. All authors have read and agreed to the published version of the manuscript.

**Funding:** This research received no external funding.

**Institutional Review Board Statement:** Not applicable.

**Informed Consent Statement:** Not applicable.

**Data Availability Statement:** Not applicable.

**Acknowledgments:** We thank Steve Bruce and Maruša Novak for comments on previous versions of this paper. We thank Maruša Novak for helping with the editing of the paper.

**Conflicts of Interest:** The author declares no conflict of interest.

## Notes

[1]   We define a mechanism as a typical causal relationship in one or several social systems (Hedström 2005; Stolz 2016).

[2]   For our purposes, a religion consists of: (1) an ideology referring to a transcendent (i.e., supernatural) reality; (2) a social group or groups producing and transmitting this ideology; (3) the individual experiences, beliefs, and actions referring to the ideology. Religiosity subsumes individual experiences, beliefs, and actions belonging to one or several religions. Examples of individual religiosity as defined here include attending religious services or a meditation course, praying, going on a pilgrimage, and believing in angels (Stolz 2020a, p. 301).

[3]   Whether we should conceive of one modernity or several modernities is a legitimate question. The idea of multiple modernities was proposed by Eisenstadt (1999, p. 284). Modernization takes on different forms in different countries and regions. In our definition of modernization, we said that modernization is a process that leads societies to a level of greater technological, institutional, and cultural complexity. It is then easy to imagine that modernization could involve all three areas (technological, institutional, and cultural) in one country while being concentrated in only one or two in another. All sorts of dosages and variations of the different elements are imaginable. Depending on the level of abstraction chosen, one might then consider modernization as "one" or "many". For our purposes here, we take modernization as a single phenomenon and focus on the varying counter-movements and mechanisms.

[4]   This part of the manuscript extends the arguments made in Stolz et al. (2023).

[5]   This point is disputed among secularization theorists, with some arguing that modernization and secularization are irreversible.

6　　The post-communist traditionally Catholic and Protestant countries do not show the same religious revival as the post-communist traditionally Orthodox and Muslim countries (except, to some extent, Croatia). The rise of Nazism in the early 1930s in Germany, following an extreme economic and political crisis, may also be an example of a form of demodernization leading to religious revival (if Nazism is seen as some kind of quasi-religion).

7　　Statistical data to demonstrate this return of the Islamic religion (religious beliefs of students) are limited, but an example given by Carvalho concerns the beliefs among students at the University of Ankara in 1978 and 1991. It is clear that belief in God and various other beliefs have increased.

8　　On the signaling aspect of the veil, see Aksoy and Gambetta (2016).

9　　Recently, data from the Arab Barometer published in the Economist showed signs of insipient secularization in six countries: Algeria, Egypt, Tunisia, Jordan, Iraq, and Libya. Between 2012 and 2019, trust in Islamic parties and religious leaders declined, the percentage of people attending mosques decreased, and the percentage of people self-identifying as non-religious increased. It is likely too early to say whether this is an indication of genuine secularization, but the finding is interesting, as some authors have argued that Arab countries are "naturally" resistant to secularization.

10　　While the Islamic revival of the 1970s was primarily caused by "too rapid modernization" and "threat from outside", there was also an important economic component to it (Carvalho 2009). According to this account, by 1970, rapid modernization from above had created a well-educated middle class. These individuals, however, did not find suitable jobs and were disappointed in their aspirations. As a result, they became open to a religious ideology arguing that self-worth was to be found not in educational and occupational prestige but in religious practice and faith (Glain 2004).

11　　The rational choice approach to religion, as represented by Stark and Finke (2000) and Iannaccone (1995), has built a whole theory around this possible mechanism, thereby likely exaggerating its importance. For some of the criticism, see Bruce (1999).

12　　We acknowledge the tremendous inequalities created during this process.

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
