# Peer review of "Explaining Religious Revival in the Context of Long-Term Secularization"

_religions, doi:10.3390/rel14060723_

Round 1
Reviewer 1 Report
The paper seeks to explain why secularisation does not happen. The author seems to believe that 'rationally' secularisation is the way that societies go when there is 'sufficient' but not 'too rapid' modernisation. How we judge whether 'too rapid' modernisation is occurring is linked to the level of religiosity in a society: when it is 'too rapid' people may resort/return to religion to help them deal with the unwelcome effects. In addition, the author lists seven more factors which, s/he argues, may encourage people to resort/return to religion: 'societal crisis', 'threat by an external power', 'introduction of supportive state regulation', 'removal of suppressive state regulation', 'internal secularisation', 'migration', 'fertility and mortality'. The author then proceeds to introduce each factor in turn, providing examples to justify the claim that such factors lead to a resort/return to religion. However, there is nothing like the required level/rigour of analysis present to enable the author to draw the conclusions s/he does with confidence. (In this respect, the work of Jonathan Fox - who the author does cite - is exemplary, enabling Fox to make conclusions about secularisation which, unlike the present paper, are underpinned by rigorous and reliable data.)
The paper is very theoretical - justified by the claim that the author is introducing a model to explain why secularisation does not necessarily occur in the way that prominent secularisation theorists such as Steve Bruce once argued would be the case. (It is noticeable that the Wallis/Bruce source much cited by the author is from three decades ago, and it is unlikely that the authors would make the same arguments today with three decades of experience to learn from.) The author makes no real attempt to measure the effects of what s/he claims to see as causes of religious 'resurgence' or 'revival'. How, for example, do we recognise 'societal crisis' when it occurs? What about 'too rapid' modernisation compared to, say, 'not too rapid' modernisation? Overall, the paper is elegantly argued, evidently written by someone who is one of the vanishing breed of secularisation theorists. Many scholars interested in secularisation would accept that secularisation is by no means a process of linear progression whichever society one looks at. Many would also recognise that trying to make a clear division between 'religious' and 'secular' in individuals, communities, nations, and so on, is problematic. It seems clear that many people combine elements of both religiosity and modernity in their psyches and to argue that it requires external (to the individual) events to make one happen faster than the other seems unfeasible.
Author Response
We thank reviewer 2 for his/her reactions. S/he makes in our view the following four points:
(1) The paper does not empirically prove its claims.
We agree, and this is also clearly stated in the paper.
(2) The paper leaves questions of empirical operationalization unsolved, for example how one should measure the strength of a "crisis" or "too rapid modernization".
Again, we agree. The ideas put forward in our paper are only useful insofar as they can be empirically substantiated, as is said in the paper. We do not think that it is useful to already go into the specifics of operationalization in such a theoretical paper.
(3) The paper assumes that secularization is a general linear process in all societies. This is an outdated perspective only held by a "vanishing breed of secularisation theorists".
Reviewer 2 is right in his/her assessment that, in our view, secularization theory claims that modernization will lead to secularization in all societies. Contrary to reviewer 2, we do not think that the theory is outdated. Our paper does not show the empirical evidence to prove our point but gives a typology of mechanisms that may explain temporary religious revivals in a context of long-term-secularization.
(4) The paper assumes that one can clearly distinguish the "religious" from the "secular". This, according to reviewer 2, is problematic.
We have seen this argument many times in the literature, but just cannot understand why it is made. Any quantitative claim about religious change will have to measure religiosity on a continuum (from very religious to very secular). Such a measurement presupposes the distinction of "religious" and "secular". If this is disputed, by definition, we cannot make any claims about religious change.
Reviewer 2 Report
The central question of this article is : How can secularization theory account the religious resurgence alongside modernization? The authors build on Wallis and Bruce typology to present eight mechanisms that can explain this revival.
The question and the scholarship on which the paper is build are obsolet. There is a lot of work that in the last two decades has shown that religion has not been erased by modernization while the secularization paradigm has been reassessed and revised :Peter Berger, Jose Casanova, Charles Taylor and Talal Asad come to mind. Casanova and Taylor are referenced in the bibliography but not really discussed in the paper.
This scholarship has brought evidence and new theories on how and why religion has n not evinced by secularization and modernization. It also shows the evolution of secularism itself, that now has to deal with relativism and plurality of belief systems, something the authors do not address.
Along the same line, no credible scholarship would advance an apriori definition of religion and focus on the “supranatural” feature.As a consequence, the exercise of identifying mechanisms is not built on solid expertise.
Additionally, I understand that the goal of the article is theoretical but the description of mechanisms has to be relevant to some empirical reality. Very vague and broad references to some situations are not sufficient.
The terminology is also ambiguous:
Revival is not really defined.
Religosity and revival of religion are not synonymous.
What is re-islamization of Turkey? What does it entail?
There are also some very broad assertions that do not reflect the reality : the Muslim Brothers movement is not a revival movement of the 1970s: it is in fact the outcome of reformist, modernist movements of the 19th century.
In sum, the paper assumes an opposition between modernization and religion, something not accepted anymore in the field of secular and religious studies.I therefore does not provide any added value to the current scholarship.
Author Response
We thank reviewer 3 for his/her reactions. Obviously reviewer 3 does not share our perspective on secularization.
Reviewer 3 states that (1) recent scholarship has shown that religion has not been erased by modernization; (2) the secularization paradigm has been reassessed and revised by scholars like Peter Berger, Jose Casanova, Charles Taylor or Talal Asad; (3) there has been an evolution of secularism itself, that now has to deal with relativism and plurality of belief systems, something the authors do not address; (4) no credible scholarship would advance an apriori definition of religion and focus on the “supranatural” feature. As a consequence, the exercise of identifying mechanisms is not built on solid expertise; (5) rvival is not really defined. Religosity and revival of religion are not synonymous. (6) it remains unclear what the re-islamization of Turkey means; (7) the Muslim Brothers movement is not a revival movement of the 1970s: it is in fact the outcome of reformist, modernist movements of the 19th century.
Our response is as follows:
ad (1) Of course, modernization has not erased religion. But quantitative evidence has shown that the level of modernization of a country is strongly correlated with the level of secularization (see the work for example by Norris/Inglehart or Molteni).
ad (2) The authors mentioned by reviewer 3 have indeed written on the secularization paradigm. But in our view, there is much empirical evidence to support sticking with the "orthodox model of secularization".
ad (3) There is indeed a lot of literature on secularism, but this is not the subject of our paper.
ad (4) Contrary to reviewer 3, we believe that scholars working on religion should use definitions as working tools. Using "supernatural" (or "transcendent") as a criterion seems to be a very useful move since the omission of this criterion leads to extremely broad definitions.
ad (5) Revival is defined in the paper. Reacting to reviewer 3's comment, we have added a definition of religion and religiosity.
ad (6) The origin of the Muslim Brotherhood is correctly stated in the paper.
Reviewer 3 Report
The main idea of this manuscript is very interesting and original. The author develops a “typology of mechanisms in which religious revival (or persistence) may occur alongside modernization.” (lines 116-17) This theory should not be seen as a revision of the secularization theory, but as a complement to it. Thus, these eight mechanisms are meant to explain the persistence or revival of religion despite ongoing modernization. (555-56)
The reconstruction of these mechanisms is placed on the broader context of what the author calls “the theory of religious – secular competition”. The basic idea is that structural changes (“macro-“) in the social, political, institutional, and technological conditions generate a process of antagonism between religious and secular actors/suppliers regarding the distribution of resources to their main clients, i.e. families and individuals (“micro-”). This is a well-written and readable article. The general structure of the paper is methodologically adequate and the reader follows easily the development of the main argument. Moreover, the author provides a vast amount of historical and sociological evidence to corroborate the development of the main argument from an empirical point of view. The overall argument is very clear and the analysis convincing. I’m just wondering if the main argument is tacitly based on a too “passive” account of religious actors, where the only thing religion can do is passively respond and react to external conditions. The Abstract should be longer and elaborate in detail the main methodological premises of the paper (the concept of competition is missing from the Abstract).
Author Response
Thank you for the appreciation.
Round 2
Reviewer 1 Report
The authors have done their best to respond to my comments on the first version of the paper. While I do not agree with all of their arguments, I think they have now done enough in terms of revisions to make the paper publishable. I am happy to see it published now.
Reviewer 2 Report
The authors have simply deflected my questions. As a matter of fact, their responses are the “by the book” example on how to not take into account feedback from reviewers.
What follows is my reaction to the authors response (in italic)
Response of the authors :Of course, modernization has not erased religion. But quantitative evidence has shown that the level of modernization of a country is strongly correlated with the level of secularization (see the work for example by Norris/Inglehart or Molteni). Norris/Inglehart perspective on secularization/modernization has been vastly criticized for one main reason: they are concerned with the individual level of secularization and do not address the institutional and societal aspects. For the institutional level and the societal level, surveys (Fox,Davie, Berger ) have invalidated their position. No credible paper on secularization, selects what is convenient for the authors and dismiss the rest.
Response: The authors mentioned by reviewer 3 have indeed written on the secularization paradigm. But in our view, there is much empirical evidence to support sticking with the "orthodox model of secularization". There is no empirical evidence in the paper that supports the “orthodox” model, if there is such a thing). In fact, the empirical evidence show the opposite of what the paper tries to advance (Fox, Driessen, Kunkler, Berger).
Response: There is indeed a lot of literature on secularism, but this is not the subject of our paper.
This one is my favorite: If the authors do not see the need to address the overlap between secularism and secularization, there is nothing academically credible in the paper. As a matter of fact, there is no strong empirical work validating the assertions advanced in the paper. The examples are anecdotes and the typology do not reflect or explain any empirical reality.
Respone: “Contrary to reviewer 3, we believe that scholars working on religion should use definitions as working tools. Using "supernatural" (or "transcendent") as a criterion seems to be a very useful move since the omission of this criterion leads to extremely broad definitions”.
What a normative position: I believe, it should etc..Again, a research paper takes into account the debates in the field in order to build an argument. No credible scholarship today starts with a definition of religion. Additionally the so called needed definition of religion is actually not connected nor operationalized to the typologies presented in the paper.
Lets’ keep in mind that for a lot of religions, the supernatural and transcendent do not apply and they are today some of the most engaged in the public space (Hinduism comes to mind).
Response: Revival is defined in the paper. Reacting to reviewer 3's comment, we have added a definition of religion and religiosity.
For which purpose? Since this definition is not connected to the argument.
Response: The origin of the Muslim Brotherhood is correctly stated in the paper.
Another response, that shows complete misunderstanding of secularization : of course the facts are correct but that was not what my feedback was about. In fact, the MB is actually a counter example of the thesis of the paper: they are NOT anti modern : they are the outcome of the modernization processes in Muslim countries. The authors’ response confuses anti modernization with anti westernization. It shows no awareness of the huge work on Islamism accumulated for half a century that has now debunked the idea that Islamism is anti modern. They are modern but not modernist: another nuance that the authors do not grasp.